# Intelligent Short-Term Multiscale Prediction of Parking Space Availability Using an Attention-Enhanced Temporal Convolutional Network

Ke Shang , Zeyu Wan, Yulin Zhang, Zhiwei Cui, Zihan Zhang , Chenchen Jiang and Feizhou Zhang *

Institute of Remote Sensing and Geographic Information System, School of Earth and Space Sciences, Peking University, Beijing 100871, China; shangke@stu.pku.edu.cn (K.S.); 2201210143@stu.pku.edu.cn (Z.W.); zhangyulin@stu.pku.edu.cn (Y.Z.); 2101210113@stu.pku.edu.cn (Z.C.); zzh_cytus@pku.edu.cn (Z.Z.); jiangchenchen@stu.pku.edu.cn (C.J.)
* Correspondence: zhangfz@pku.edu.cn

**Abstract:** The accurate and rapid prediction of parking availability is helpful for improving parking efficiency and to optimize traffic systems. However, previous studies have suffered from limited training sample sizes and a lack of thorough investigation into the correlations among the factors affecting parking availability. The purpose of this study is to explore a prediction method that can account for multiple factors. Firstly, a dynamic prediction method based on a temporal convolutional network (TCN) model was confirmed to be efficient for ultra-short-term parking availability with an accuracy of 0.96 MSE. Then, an attention-enhanced TCN (A-TCN) model based on spatial attention modules was proposed. This model integrates multiple factors, including related dates, extreme weather, and human control, to predict the daily congestion index of parking lots in the short term, with a prediction period of up to one month. Experimental results on real data demonstrate that the MSE of A-TCN is 0.0061, exhibiting better training efficiency and prediction accuracy than a traditional TCN for the short-term prediction time scale.

**Keywords:** parking space availability prediction; multi-time scale; intelligent transportation; temporal convolutional network; deep learning

## 1. Introduction

The provision of sufficient places for public needs has an impact on sustainable development and effective spatial planning when creating a Smart City [1–3]. Accurately predicting parking space availability is a vital component of parking guidance information (PGI) systems. In medium- to large-sized cities, parking scarcity often forces users to expend valuable time and energy searching for parking on city-level and on-site roads. Parking cruises account for an average of 30–40% [4] and drivers spend 3.5–14 minutes searching for roadside parking spaces. In large cities such as Chicago, people might drive 172 million miles annually to find parking spaces, owing to unknown information about parking availability. This is equivalent to 8.37 million gallons of gasoline consumption and over 129,000 tons of carbon dioxide emissions [5]. Disorderly and inefficient traffic flow can disrupt normal traffic patterns and negatively impact the overall transportation network. Users' parking decisions are based on time and are influenced by multiple prior knowledge and current experience. Through the dynamic updating of parking place information and accurate prediction, users and administrators can make informed decisions about the traffic flow in parking lots. This promotes efficient and user-friendly parking guidance and increases the utilization of parking places in lots and on adjacent roads [6], promoting the balanced utilization of parking facilities [7].

Owing to the high complexity, randomness, and uncertainty of urban parking behavior, it is challenging to predict the short-term availability of parking places. As the

prediction time interval is further shortened, the level of uncertainty regarding parking place availability increases, making it progressively challenging to design a mathematically precise model. Therefore, the use of intelligent prediction techniques and the fusion of multiple prediction models are effective approaches and development trends for improving the accuracy of short-term parking space prediction.

## 1.1. Literature Review

The availability of parking places in parking lots is characterized by temporal fluctuations, spatial non-uniformity, and limited resources. Adopting appropriate methods to predict parking space availability in real time and with precision can lead to the more rational utilization of parking resources [8]. Time series of parking place occupancy have the following characteristics: (1) nonlinear relationships exist among various influencing factors; and (2) they exhibit stochastic fluctuations, meaning that if the number of data points is small and the data fluctuates significantly over a short period of time, some neural network models may be trapped in local minima, leading to larger prediction errors [8]. Currently, research on parking space prediction focuses mainly on its temporal characteristics, and extracts and seeks to predict its features through time series forecasting algorithms.

Earlier methods for predicting parking place availability include linear time series forecasting methods [9,10], which convert all influencing factors into time factors and assume linear relationships between variables in the system, allowing for/enabling fast prediction results. However, owing to the neglect of nonlinear factors, the accuracy of the prediction results is relatively low. To address the non-negligible issue of nonlinearity, various nonlinear time series prediction methods, such as chaotic time series [11], autoregressive moving average models [12], and exponential smoothing methods, have been used. However, these methods still have some limitations in practical applications due to their theoretical complexity, the necessity for significant human involvement in model analysis, and the need for enhanced prediction accuracy.

Since the emergence of machine learning theory, neural network models have gained attention for their strong fault tolerance and robustness, as well as their ability to identify nonlinear and complex systems, making them suitable for predicting parking space availability [13,14]. In recent years, various parking space availability prediction models using neural network analysis have been developed, including parking place prediction models based on fuzzy neural networks and back-propagation (BP) neural networks [15,16]. Traditional BP neural networks assume, without theoretical justification, that future changes in parking space data are determined by the parking space data from previous time intervals. This approach is somewhat arbitrary and leads to inaccurate prediction results. Parking prediction models based on wavelet neural networks [17] and those based on wavelet transforms and particle swarm wavelet neural networks [18] use wavelet transforms to analyze input data and then use wavelet neural networks to predict parking space availability, reducing the interference caused by initial data fluctuations [8] and improving prediction accuracy. However, wavelet transforms require a significant amount of time to process, which affects their efficiency.

Currently, the theme of sequence modeling under the background of deep learning is mainly related to recurrent neural networks (RNNs), such as long short-term memory (LSTM) and the gated recurrent unit (GRU). LSTM can model long-term dependencies and solve the problem of gradient disappearance through memory cells and gating functions. The LSTM neural network has significant advantages in time series prediction [19] and has been applied in fields such as stock prediction in financial markets and short-term traffic flow prediction [20–22]. In [23], parking vacancy rates on the street and the likelihood of cars leaving were used as efficiency indicators for prediction. An LSTM model was used for spatiotemporal analysis of the parking place occupancy rate. The experiment showed that the accuracy of network prediction was improved by incorporating spatial correlations, with a mean averaged error (MAE) of 0.018. In [24], researchers proposed a solution for predicting traffic flow using a combination of stacked autoencoders (SAEs)

and LSTM. LSTM is used to capture temporal features while SAE is used to extract spatial features. In [25], LSTM was used to predict the number of available places (NAP), achieving a model accuracy with a root mean squared error (RMSE) of 5.42. In [26], an RNN was used to simulate parking occupancy rate, achieving an MAE of 0.067. However, the standard LSTM neural network used for time series prediction has the disadvantages of long computation time and high complexity [22]. In many reports, bad weather tends to have a negative impact on traffic. Vlahogianni et al. [27] have pointed out that real-time traffic flow prediction is critical to traffic management systems during these kinds of situations, especially heavy rain or rainstorms. Based on the multivariate regression function and the Markov model, Tanimura et al. [28] and Li et al. [29] proposed prediction methods of traffic flow on snow roads. Others describe rain, snow, fog and other weather as bad weather. Using a random forest method, an SVR model and a deep belief network, a traffic flow prediction model for bad weather was built [30,31]. Neural networks, such as GRU, RNN, and LSTM [32,33], can take weather parameters as input data. It has been found that the introduction of weather data can improve the accuracy of traffic flow prediction [34].

### 1.2. Research Questions

With our proposed approach, we aimed to address three key questions:

(a)  Is there an alternative calculation method that is faster and more efficient than existing methods? Can this method effectively predict parking space availability, by considering it as a time series with periodic changes?

(b)  Which factors should be considered? Multiple influencing factors, such as dates, weather conditions, etc., are usually expressed in textual form. How can these factors be quantified and incorporated into numerical calculations?

(c)  What is the appropriate time scale for prediction? Should the prediction time interval be in minutes, days, or months?

### 1.3. Proposed Solution

To provide clear and concise answers to these questions, we chose a method based on the TCN network model to assess parking space availability.

(a)  Firstly, the applicability of TCN was evaluated using a single input of time series. Then, the parameterization of multiple influencing factors was discussed.

(b)  To evaluate the impact of various parameters on parking space availability, the network was improved by incorporating multiple input channels and implementing attention mechanisms to achieve an efficient allocation of information processing resources.

(c)  We expected the time precision of prediction to be as refined as possible without wasting computational resources, to provide reference for precise regulation of traffic flow. However, longer periodic predictions can provide longer preparation time for the formulation and implementation of residents' travel and transportation policies from a macro perspective, similar to weather forecasts. Therefore, we considered two time scales: ultra-short-term predictions with minute intervals and short-term predictions with daily intervals. We validated the prediction results of available parking spaces using these two scales, and then evaluated the prediction of the parking lot congestion index by considering multiple influencing factors.

### 1.4. Research Contributions

Our major contribution is proposal of an attention-enhanced TCN—A-TCN—with an added spatial attention module (SAM) used to preprocess the input data to effectively enhance the network training efficiency, shorten the learning time, and accelerate the convergence speed while improving the prediction accuracy. Simultaneously, the parking place prediction is divided according to the time scale into ultra-short-term dynamic prediction of the NAP, using minutes as the unit of time, and short-term prediction, using days as the unit of time. The busy index (BI) is proposed as the evaluation index for

short-term prediction, and multiple influencing factors, such as associated dates, extreme weather, and human regulation, are comprehensively considered. Using real parking place time series records from a comprehensive commercial district underground parking lot in the year 2021, the proposed A-TCN was trained, and ultimately achieved a precision, given by a mean squared error (MSE), of 0.96 and 0.0061 on the ultra-short-term and short-term time scales, respectively. The main innovation points of the paper are as follows:

(a)  This paper proposes different TCN solutions for different time units and time sequence lengths in various scenarios. For ultra-short-term NAP prediction, data at each minute of a single natural month were used for the dataset. Unlike previously reported approaches that excluded static data at night, this study retains the continuity of time-series features, which leads to higher accuracy. This suggests that the continuity of static data as a sequence in the time dimension should not be ignored, as it plays an important role in the network's learning and optimization process. For short-term parking predictions on a daily basis, the BI was introduced to quantify the availability of parking places in the parking lot. The impact of various factors, such as date, extreme weather, and human regulation, was considered to forecast time-series changes over a relatively longer time span, which represents a novel approach.

(b)  To achieve the described functions, a network architecture for A-TCN is proposed based on the original TCN. To address the influence of multiple factors, a spatial attention mechanism module is introduced to intelligently adjust the weights of different factors without changing the size of the input tensor. For datasets with different amounts of data, suitable network depths and training parameter suggestions are provided. The obtained accuracy results are intended to be higher than those achieved in previous studies.

## 2. Materials and Methods

According to some researchers [35], when modeling sequential data, convolutional neural networks (CNN) can achieve better performance than RNNs while avoiding the common flaws of recursive models such as gradient explosion, vanishing, or lack of memory retention. In addition, the use of a CNN instead of an RNN can improve performance because it allows for parallel computation of outputs. From this basis, the temporal convolutional network (TCN) has emerged, firstly used for video-based action segmentation [36], and later used for time series prediction such as weather forecasting. The TCN has performed better than LSTM with the following advantages:

(a)  Parallelism: TCN allows for a higher speed compared with RNN.
(b)  Flexible receptive field size: TCN provides better control over the model's memory size and adaptability to different domains.
(c)  Stable gradients: TCN avoids the issues of gradient explosion/vanishing that are commonly encountered in RNN networks.
(d)  Low memory requirements: TCN exhibits lower memory requirements during training.
(e)  Variable-length inputs: TCN can handle sequence data of any length, providing flexibility in dealing with inputs of different sizes.

Based on these advantages, we have considered the use of TCN as the basic architecture of the network model. To effectively enable the network to balance multiple input channels and handle multiple factors, a spatial attention module is added to the front end of each input to form an improved A-TCN network. It is necessary to preprocess the input data, including time scale normalization of time series, parameterization of influencing factors, etc. These preprocessing steps ensure that all original data are unified into a time series before being used as input to the network.

### 2.1. Attention-Temporal Convolution Network

2.1.1. Network Architecture

The A-TCN model was improved by using causal convolution to adapt to sequence models. Dilated convolution and residual blocks are used to capture historical memory.

Because the focus of the research is on time series data, the A-TCN model, similarly to TCN, adopts one-dimensional convolutional networks. As shown in Figure 1, the A-TCN architecture uses causal convolution and dilated convolution. The model built consists of three hidden layers, one input layer, and one output layer. Kernel size is set to 2, that is, the input of each layer is the output of the previous layer at two moments. Dilation parameters are set as (1, 2, 4, 8), indicating the time interval between the inputs of each layer. Dilation = 4 means the output of the previous layer is shifted forward by 4 time-steps to serve as the input of the current layer, until enough inputs (based on kernel size) are taken. The same principle applies to the dilations of 1, 2 and 8. To achieve time series prediction, TCN utilizes a 1-D FCN structure, where each hidden layer has the same input and output time lengths and maintains the same time steps. To address the second challenge of handling long-history information, TCN uses dilated causal convolutions.

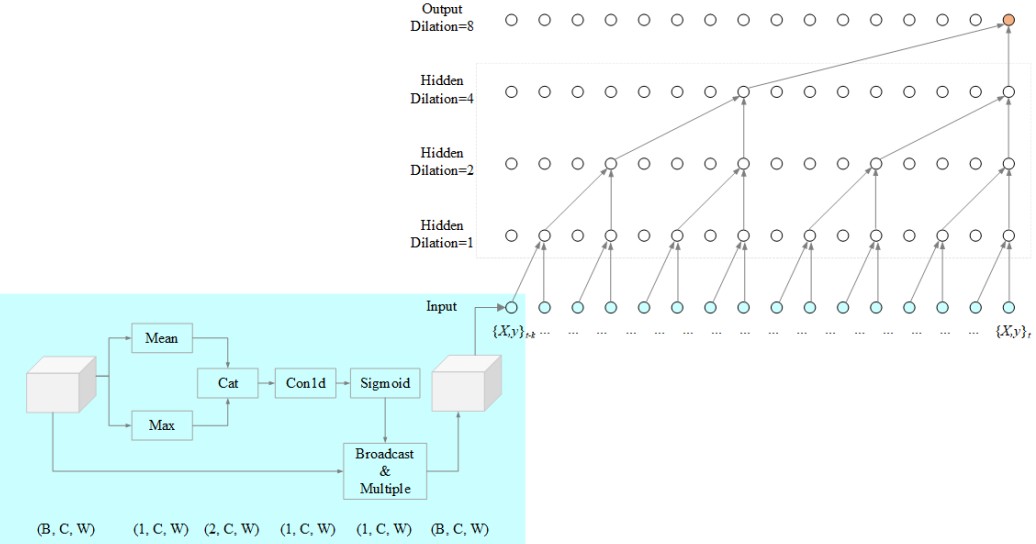

**Figure 1.** Network architecture of an attention-enhanced temporal convolutional network (A-TCN).

At each layer, the value at time t depends only on the values of the previous layer at time t, t − 1, etc., reflecting the characteristics of causal convolution. The information extraction at each layer from the previous layer is undertaken in a skip-connection manner, and the dilation rate increases exponentially by a factor of 2, reflecting the characteristics of dilated convolution. Because of the use of dilated convolutions, padding (usually filled with 0) is required for each layer, with the padding size being (k − 1)d.

2.1.2. Inputs and Targets

The training set is composed of pairs of equal-sized sub-sequences of the given time series (input sequence, target sequence), that is, input series = target series, where the target sequence is the sequence that is shifted to the right by a certain output_length relative to its respective input sequence, as shown in Figure 2. This means that the target sequence of length input_length contains the last (input_length–output_length) elements of its respective input sequence as its first element, and the output_length elements after the last entry of the input sequence as its last element. In terms of prediction, this means that the maximum prediction horizon of the model is equal to output_length. Using a sliding window approach, many overlapping input and target sequences can be created to form a time series.

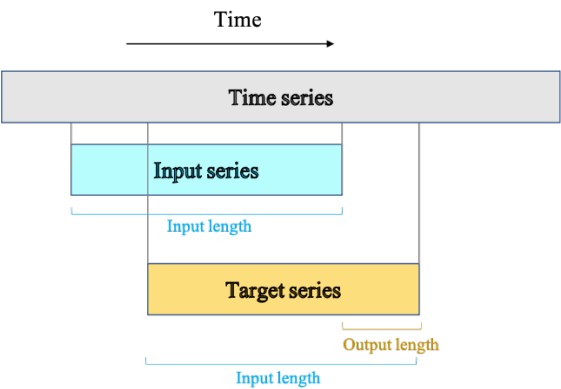

**Figure 2.** Prediction method of A-TCN.

### 2.1.3. Data Preprocessing Module

In short-term forecasting tasks with a yearly span and a daily time sampling unit, factors such as holidays, extreme weather conditions, and policies can have a significant impact on parking space availability. Considering that these factors may also contain certain time patterns, such as the periodicity of holidays and climate, adding them to the output end of the network could result in the loss of this information. Therefore, we integrated them into the input end of the network.

As shown in Figure 3, complex patterns and the BI have the same dimension (time, 1). They are merged into a tensor of dimension (time, num) based on the second dimension. Then, according to the setting of window_size, the data are shifted and concatenated into a temporal sequence, and then cropped to obtain a tensor of dimension (time, num, window_size). Finally, the dataset is divided into a training set and a validation set with a ratio of 4:1, which is input into the network.

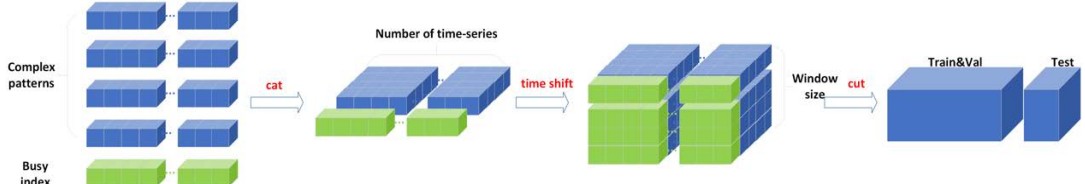

**Figure 3.** Data preprocessing.

### 2.1.4. Spatial Attention Module

The SAM is shown in Figure 4 and takes the feature map F' output from the channel attention module as its input feature map. First, a channel-based global max pooling and a global average pooling are performed to obtain two H × W × 1 feature maps. Then, the two feature maps are concatenated along the channel dimensions. After that, a 7 × 7 convolution operation is applied to reduce the dimensionality to one channel of H × W × 1. The sigmoid function is applied to generate the spatial attention feature $M_s$. Finally, the input feature and the spatial attention feature are multiplied together to obtain the final generated feature.

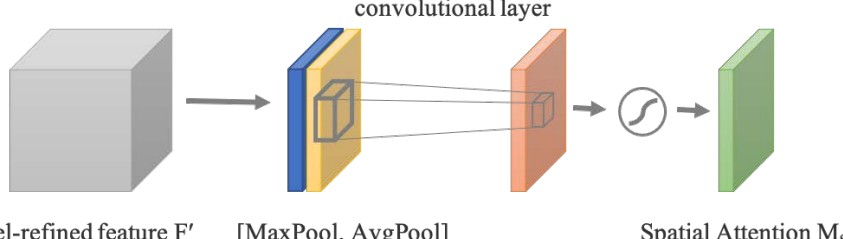

**Figure 4.** Spatial attention module (SAM).

### 2.2. Data Preprocessing Method

The DS1.0 multivariate dataset consists of the NAP time series for the entire year of 2021, as well as various factors that influence it, such as meteorological data, associated dates, and autonomous intervention information, as shown in Table 1. The preprocessing of the multivariate data includes parameterization and normalization of the time dimension. Specifically, the NAP data are processed as follows: a dynamic time series of NAP is generated using natural minutes as the unit, the BI is calculated on a daily basis using days as the unit, and other influencing factors are considered.

**Table 1.** Data source information.

| Time Series Type | Time Interval | Start Date | End Date |
|---|---|---|---|
| Available parking places | 1 min | 1 January 2021 | 31 December 2021 |
| Multivariate factors | 1 day | | |

#### 2.2.1. Normalization of NAP Time Series

The dataset was obtained from a large comprehensive commercial underground parking lot where users consist of employees from surrounding business offices, shopping mall consumers, and a small number of other users. The data were collected from 1 January to 31 December in 2021, and the time interval was normalized to minutes. The total length of the obtained time series is 525,600 points.

The historical available parking space time series is shown in Figure 5. There were abnormal changes in the available parking place data during certain periods of time, for instance, 11 February was Chinese New Year's Eve, extreme weather occurred from 20 to 22 July, the parking lot was affected by extreme weather from 20 July to 29 August, and the parking lot was closed on 22 August.

The BI is calculated based on the proportion of the total duration of congestion that occurred during a natural day, where congestion is defined as the time when the NAP in the parking lot is less than 10% of the total capacity. The calculations are shown in (1) and (2).

$$I_b = \frac{\sum t_b}{T_d} \tag{1}$$

$$t_b = t|_{n_a < n_{limit}} \tag{2}$$

where $I_b$ is the BI, $t_b$ is the duration when the NAP is less than the threshold value, $T_d$ is the total time that the parking lot is open each day, $n_a$ is the current NAP, and $n_{limit}$ is the threshold value for congestion.

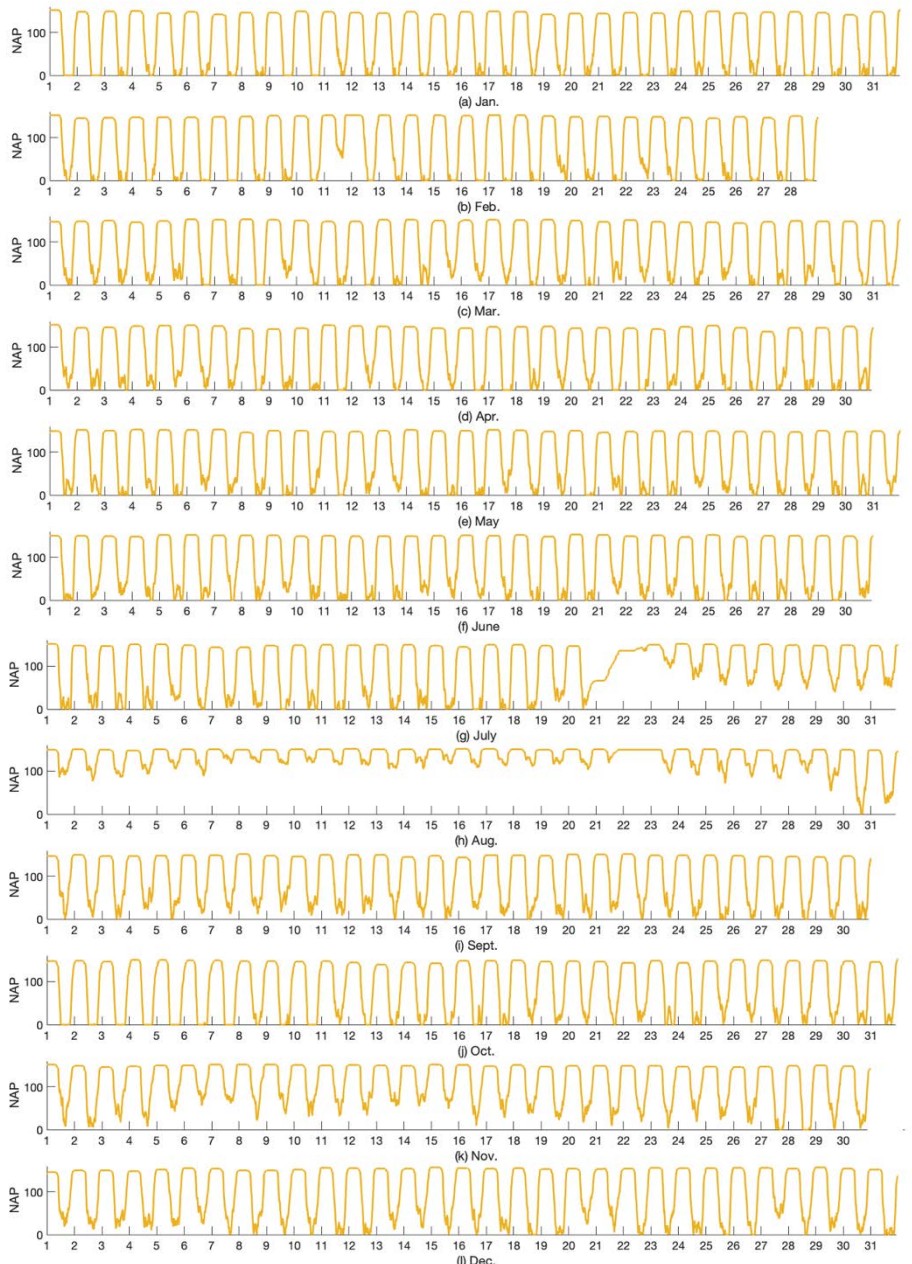

**Figure 5.** Dynamic variation curve of number of available places (NAP) in the dataset.

2.2.2. Multidimensional Evaluation and Quantification of Influencing Factors

In [23], researchers evaluated the influence of three factors—day of the week, occupancy rate and the time of the day—on the duration time. For comprehensive parking lots, the dynamic changes of parking places are related to user attributes and relevant influencing factors. The parking behavior of users shows changing patterns related to the following factors: working days vs. holidays, extreme weather conditions, and whether there is human control in parking induction. Therefore, this paper quantifies these factors separately as the date index (DI), weather index (WI), and control index (CI) defined as follows:

1. DI: The associated dates are classified into three types: working days not adjacent to holidays, working days adjacent to holidays, and holidays. The weight of working days not adjacent to holidays is 0, the weight of working days adjacent to holidays is 0.05, and the weight of holidays is 0.1.

2. WI: Ordinary weather does not have a significant impact on parking space availability in the city's parking lots, so only the suppression effect of extreme weather conditions is considered. The parameter weight of extreme weather is set to 0.1, and the weight of common weather is 0.

3. CI: Negative interventions, such as traffic control, are the primary consideration, mainly manifested as the release of parking resources. The weight during such periods is set as 0.1, and the weight for periods with no intervention is set as 0.

Table 2 lists the influencing factors and weights used in this study. The multiple influencing factors are plotted as a scatter plot in Figure 6.

**Table 2.** Types of influencing factors and parameter weights.

| Index | Parameter | Weight |
|---|---|---|
| DI | Working days not adjacent to holidays | 0 |
| | Working days adjacent to holidays | 0.05 |
| | Holidays | 0.1 |
| WI | Common weather | 0 |
| | Extreme weather | 0.1 |
| CI | None | 0 |
| | Suppression | 0.1 |

Note: DI, date index; WI, weather index; CI, control index.

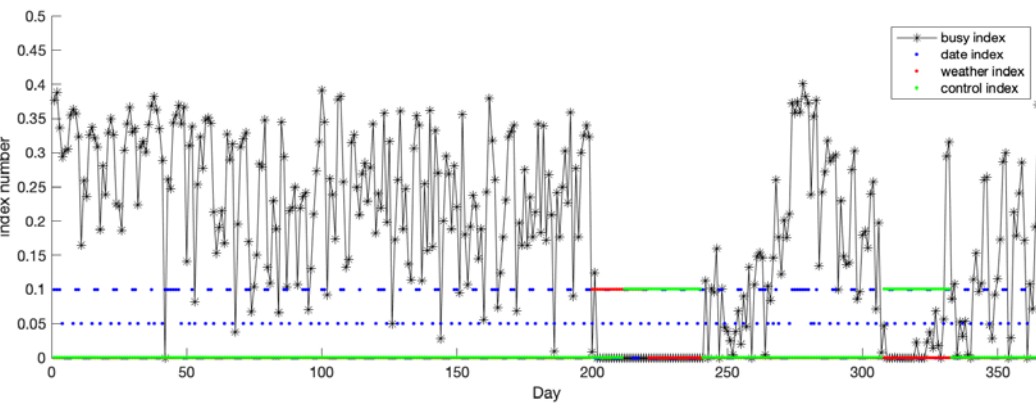

**Figure 6.** Parameter curve and scatter plot.

### 2.3. Performance Measures

The training was conducted on a computer with a 14-core Intel(R) Xeon(R) Gold 6330 CPU@2.00 GHz, an RTX 3090 GPU with 24 GB memory, and the operating environment was Python3.8 (ubuntu18.04) + PyTorch1.8.1 + Cuda11.1. The batch size was set to 30, and the number of epochs was set to 200. For this study, 5 layers were used for the network, with 30 hidden nodes in each layer. The window size was set to 10, the learning rate was set to 0.004, the drop-out rate was set to 0, and the input historical time series window length was set to 10. The data in the training set were randomly shuffled.

The loss function used to train the model was the MSE between the predicted values and the actual values. In parameter estimation, MSE refers to the expected value of the squared difference between the parameter estimate and the true value. In machine learning, MSE is usually used as a function to evaluate network performance and is also known as the L2 loss function, expressed as

$$\text{MSE} = \frac{1}{m}\sum_{i=1}^{m}(y_i - \hat{y}_i)^2 \tag{3}$$

where $y_i$ is the input data, namely, the time series in the training set; $\hat{y}_i$ is the output data, namely, the prediction of the network; and m is the total number of data points.

For ultra-short-term forecasting, three networks were trained for comparison: a TCN with traditional univariate input, a TCN network with multivariate input, and the proposed A-TCN network with multivariate input. Short-term prediction was performed on the corresponding parameter indices from day 295 to day 365 of the year 2021 in the testing set.

## 3. Experiment and Results

### 3.1. Network Training

Training, validation, and testing datasets were divided as shown in Table 3. To eliminate the correlation between data and prevent the model from overfitting to some data, the data in the training set were first randomly shuffled. The MSE between the predicted values and the actual values was then used as the loss function to train the model. Total training time was 1 h and 57 min.

**Table 3.** Dataset information (data points).

| Time Scale | Time Unit | Number of Data Points |
|---|---|---|
| Ultra-short | Per minute in one month | 44,640 |
| Short | Per day in one year | 365 |

### 3.2. Analysis of Ultra-Short-Term Forecasting Results on TCN

The MSE curves for the validation and testing sets are shown in Figures 7 and 8, respectively. After 120 epochs, the MSE reached 0.96, indicating that the network had converged.

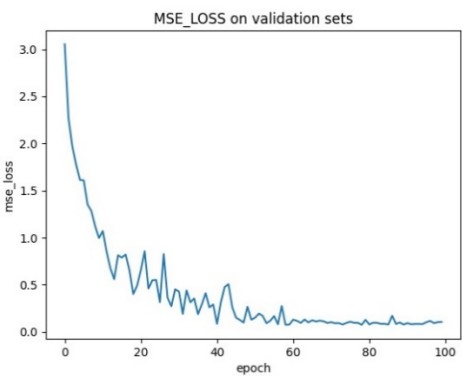

**Figure 7.** Mean squared error (MSE) curve for the validation set.

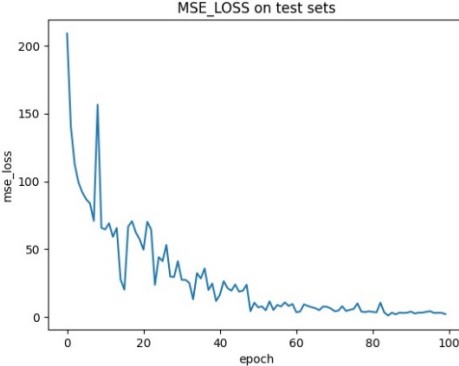

**Figure 8.** MSE curve for the testing set.

As shown in Figure 9, the predicted curve of the test set exhibited a high degree of agreement with the true values, indicating that it could effectively forecast the changes in

NAP during a day. This confirms the good performance of TCN in the ultra-short-term prediction of parking space availability.

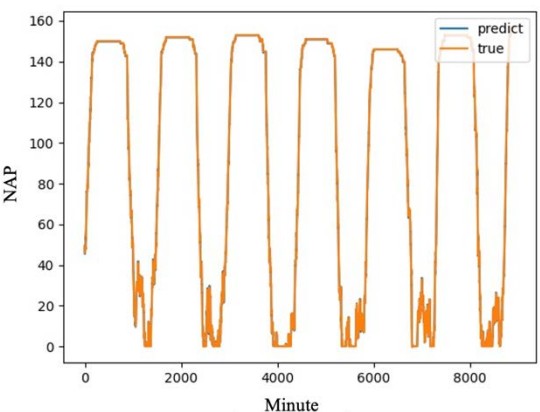

**Figure 9.** True and predicted curves of the testing set.

### 3.3. Analysis of Short-Term Forecasting Results on A-TCN

The model parameters were trained using a TCN with traditional univariate input, a TCN with multivariate input, and the proposed A-TCN with multivariate input. In the validation set, the TCN with multivariate input had the fastest error curve descent rate, but in the test set, it quickly overfitted with increasing network iteration (Figures 10 and 11). The A-TCN had a slower convergence rate, but it was less prone to overfitting and had better accuracy than TCN. Ultimately, the A-TCN achieved an MSE of 0.0061 after 155 epochs, with higher accuracy than the other two network models, as shown in Table 4. This confirms that the proposed A-TCN has better accuracy and robustness.

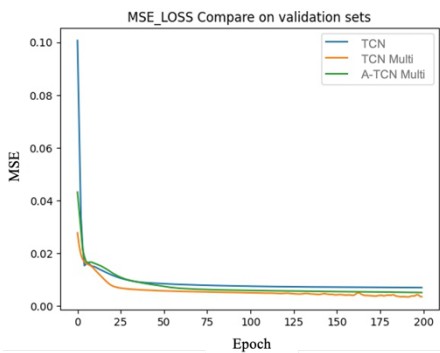

**Figure 10.** Validation results.

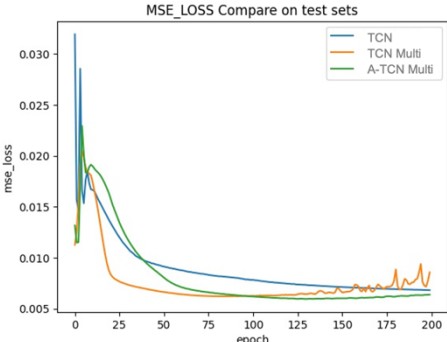

**Figure 11.** Test results.

**Table 4.** Network efficiency comparison of different methods on the DS1.0 dataset.

| Method | MSE |
| --- | --- |
| TCN with univariate input | 0.0083 |
| TCN with multivariate input | 0.0079 |
| **A-TCN with multivariate input** | **0.0061** |

Note: MSE, mean squared error; TCN, temporal convolutional network; A-TCN, attention-enhanced TCN.

Short-term prediction was performed on the corresponding parameter indices from day 295 to day 365 of the year 2021 in the testing set, and the prediction results for BI are shown in Figure 12.

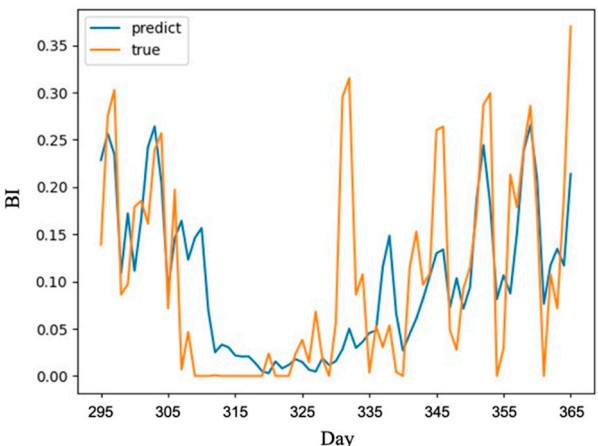

**Figure 12.** Short-term prediction results for the busy index (BI).

The overall trend of the network's prediction results matches the true values, especially for the periods where the level of congestion increased—day 296 to 297, 303 to 304, 345 to 346, 352 to 353, 358 to 359, and day 365—indicating good prediction performance.

The prediction performance was not satisfactory from day 306 to day 340, with a time delay in reflecting the abrupt drop from day 306 to day 319 and the sudden rise from day 331 and day 332.

## 4. Discussion

### 4.1. Advantages and Drawbacks of TCN

In short-term prediction, several increases and decreases of BI were worked out by the proposed A-TCN, while two changes, the abrupt drop from day 306 to day 319 and the sudden rise from day 331 and day 332, were missed. After investigation, it was found that both periods experienced spontaneous human behavior due to uncontrollable factors, which were different from the previous patterns and resulted in a sharp change in BI. However, this spontaneous factor belongs to a random variable that does not exist in the historical data, so it was not taken into consideration during network training, and the network could not provide effective feedback for this situation.

The network architecture based on the TCN has the following advantages: parallelism, flexible receptive field size, stable gradients, low memory requirements during training, and variable-length inputs. Based on these features and the results of this study, A-TCN is believed to be the most suitable basic network model currently available for the short-term prediction of parking space availability. However, although not yet discovered in current work, the network has the following potential drawbacks that may cause difficulties for large-scale prediction work:

1.  Data storage during validation. During validation/testing, an RNN only needs to maintain the hidden state and obtain the current input to generate predictions. In other words, a fixed length set of vectors provides a summary of the entire history, and the

observed sequence can be discarded. In contrast, a TCN needs to receive the original sequence of valid historical length, so it may require more memory during evaluation.

2. Potential parameter change for a transfer of domain. Different domains may have varying requirements for the number of historical data points needed for model predictions. Therefore, when transferring the model from a domain that requires minimal memory to one that requires longer memory, a TCN may not have a sufficiently large receptive field.

### 4.2. Prospects

In this study, we propose an A-TCN-based model suitable for parking availability prediction. Through an empirical example validation, the prediction model has demonstrated strong performance. In particular, this method can help drivers to predict the number of available parking spaces in a parking lot. In addition, it can assist in traffic control and reduce traffic congestion in some areas, and help parking lot management by providing predictions of parking information for future periods.

It is worth noting that we believe that the role and correlation of influencing factors vary with geographical location and parking lot characteristics. For example, parking lots in shopping malls tend to be busier and experience higher demand during holidays, whereas parking lots in office areas exhibit greater activity on weekdays. Parking lots near hospitals may be busier and greener during seasons of high disease incidence, while comprehensive parking lots perform more evenly or have more complex influencing factors. Therefore, it will be crucial to explore and discuss the importance and correlation of these influencing factors in addition to berth prediction algorithms. We hope to investigate this aspect in future research. Future studies should also consider a broader range of influencing factors, such as the positive incentives generated by policy guidance on traffic flow, the impact of main road traffic flow on parking space availability over time, spontaneous collective events, and spatially relevant influencing factors. After exploring various methods for parking space availability prediction, it is important to investigate how to regulate and manage parking to alleviate congestion and increase utilization, ultimately addressing the challenges of limited parking space availability and slow parking, and to enhance the key technologies and solutions for PGI systems.

### 5. Conclusions

The effective prediction of NAP and the rational guidance of traffic play crucial roles in optimizing the utilization of urban transportation infrastructure, reducing ineffective and disorderly traffic flow, and decreasing exhaust emissions and resource waste. In this study, we propose an A-TCN for the dynamic prediction of NAP, which performs ultra-short-term prediction of parking space availability on a minute-level time series and short-term prediction on a daily-level dataset. We propose to use BI as the evaluation index for short-term prediction, considering multidimensional influencing factors such as DI, WI, and CI.

On an annual-cycle dataset of a certain parking lot, experimental results demonstrate that the traditional TCN achieved a prediction accuracy of MSE = 0.96 for ultra-short-term forecasting and was effective in predicting the NAP sequence. For short-term forecasting, the proposed A-TCN approach achieved an accuracy of MSE = 0.0061, which exhibits significant improvement in both prediction accuracy and training efficiency compared with the TCN (with a univariate variable MSE = 0.0083 and a multiple-input variable MSE = 0.0079).

**Author Contributions:** Conceptualization, Ke Shang, Feizhou Zhang, Zeyu Wan, Yulin Zhang, Zhiwei Cui, Zihan Zhang, Chenchen Jiang; methodology, Ke Shang, Zeyu Wan, Feizhou Zhang; software, Zeyu Wan; validation, Ke Shang, Yulin Zhang, Zhiwei Cui, Zihan Zhang, Chenchen Jiang; formal analysis, Ke Shang, Yulin Zhang, Zhiwei Cui; writing—original draft preparation, Ke Shang, Zeyu Wan, Feizhou Zhang; writing—review and editing, Ke Shang, Zeyu Wan, Feizhou Zhang; visualization, Yulin Zhang, Zhiwei Cui, Zihan Zhang, Chenchen Jiang; supervision, Feizhou Zhang. All authors have read and agreed to the published version of the manuscript.

**Funding:** This research received no external funding.

**Data Availability Statement:** Data and codes are unavailable due to privacy.

**Acknowledgments:** Local government is thanked for the assistance in providing the dataset. MDPI IJGI editorial office is thanked for the help and support. Reviewers are appreciated for their suggestions of modification.

**Conflicts of Interest:** The authors declare no conflict of interest.

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
