# Peer review of "Intelligent Short-Term Multiscale Prediction of Parking Space Availability Using an Attention-Enhanced Temporal Convolutional Network"

_ijgi, doi:10.3390/ijgi12050208_

Round 1

Reviewer 1 Report

Dear Authors,

The issue raised in the manuscript is interesting and needed in the context of transport problems in cities and agglomerations. In general it conforms to the standards of scientific articles.

However, I would like to provide some comments on the content and structure of the manuscript:

11. There is no clear indication of the purpose of the research and research results in the Abstract.

22. „Introduction” is definitely too long and at the same time does not contain the necessary elements:

·       there is no clear indication of the purpose of the research in the Introduction,

·       I do not see any research questions  (RQs) in the Introduction. After adding RQs, it will be required to answer them in the content of the manuscript.

33. There is no „Literature review” part. The number of literature items is definitely too small (only 23!!!) Moreover, you mostly used only Asian research. Please remember that IJGI is an international journal. I propose to read and use also European and American literature in this field;

44. Conclussions are missing in the manuscript.

55. What are the practical aspects of the research? You need to define them.

Author Response

Dear reviewer:

Thank you for your review and constructive comments on our manuscript entitled “Intelligent Short-Term Multiscale Prediction of Parking Space Availability Using an Attention-Enhanced Temporal Convolutional Network” (ID: ijgi-2365315). Your suggestions have been thoroughly reviewed, and we have made the necessary revisions accordingly. The revised sections are marked up in blue in the manuscript. We sincerely hope that these corrections address your concerns and meet with your approval. Please find in the attachment the main corrections we made in response to your comments.

Reviewer 2 Report

First of all, I would like to say that this manuscript focuses on a very interesting research problem.

TITLE

The article’s title is suitable with the content of the paper.

ABSTRACT

The abstract is well-designed and briefly express the present research thus being of interests and readable thus capturing the reader’s attention. It present in an appropriate manner the main research hypothesis, the problem statement, the methods and the main findings.

KEY WORDS

The key words are appropriate to the present research and are clearly stated.

ORIGINALITY

The article meets a high level of originality argued by the main research theme and the research hypothesis. Furthermore, the originality of the paper is highlighted by the main results of the paper.

The authors construct a well-designed theoretical background closely related to the current specialised literature in the field. A short recommendation I would like to made, it is stated in the final part of this review form.

THE PAPER S STRUCTURE

The structure of the paper is correct in line with the journal standards and meet the publication requirements considering the paper logic. The objectives seem to be clear formulated as well as the investigation is drawn. The core argument of the paper illustrates the paper relevance and the research originality. The results are clearly express and well connected both to the theoretical framework and discussions.

The paper should be organized into the following sections: 1) Introduction; 2) Literature review; 3) Research methods; 4) Study area; 5) Results; 6) Discussion; 7) Summary.

THE METHODS

The methodological design is appropriate and the methods fit well to the present investigation. I suggest the authors add the sentence to the introduction:

Providing places for public needs has an impact on sustainable development and effective spatial planning when creating a Smart City (Ogryzek M, Krupowicz W, Sajnóg N. Public Participation as a Tool for Solving Socio-Spatial Conflicts of Smart Cities and Smart Villages in the Sustainable Transport System. Remote Sensing. 2021; 13(23), pp. 1-24, 4821. https://doi.org/10.3390/rs1323482. and Yin, C., Xiong, Z., Chen, H., Wang, J., Cooper, D., & David, B. (2015). A literature survey on smart cities. Sci. China Inf. Sci., 58(10), 1-18.) and Pereira, G. V., Parycek, P ., Falco, E., & Kleinhans, R. (2018). Smart governance in the context of smart cities: A literature review. Information Policy, 23(2), 143-162.,)

THE MAIN ANALYSIS

The main research is well design and appropriate conducted in line with the main questions in spatial planning in different European regions.  I suggest the authors to be a little more specific when disrobe a process and methods.

The text lacks considerations on the problems encountered during the work, constructive criticism of scientific methods and a description of how other scientists solved this problem. How learning is done and indication of what is automatic and what is manual process.

CONCLUSIONS

the conclusions fit well summarising the main ideas of the present analysis.

THE GRAPHICAL SUPPORT

The graphical support is well formatted, appropriate illustrating the text content.

THE ENGLISH LANGUAGE

I think the English is ok as far as I could see. I enjoyed to read this paper in English and the language seems well but I think that an opinion of a native English speaker is welcomed. In other words, if the authors used a specialised proofreading services and they could prove this aspect I trust the opinion and the work of this proofread service. On the other hand, I put my trust regarding the English language on the journal editors but I repeat the language seems well.

RECOMMENDATIONS

Finally, I recommend the publication of this paper with some minor revision considering the above mentioned aspects, references and citations.

I want to see the revised version of this paper before publication for a final acceptance and to ensure that the revision has been completely and carefully made.

Author Response

(The authors gave the same response as above.)

Reviewer 3 Report

This article addresses parking prediction availability using based on a Temporal Convolutional Network (TCN) model with some factors that influences parking availability. The language, structure, references, and experiments are well.  However, some points need to be addressed:

- the introduction is too long, I suggest splitting it into two parts, the first one is about the research problem and literature review and the other can be a new section about machine learning for car park availability prediction. 

- there are many factors influencing parking availability, you did not mention any reason for choosing only three factors in your study. 

- the importance of each influencing factor can be ranked from the highest to the lowest importance.  So that the prediction power for each factor can be determined. If possible, I suggest considering this point.

The language is fine

Author Response

(The authors gave the same response as above.)

Reviewer 4 Report

The authors are suggested to list the conclusions in an seperate Conclusion Section.

The data and code availability is not mentioned.

Author Response

(The authors gave the same response as above.)

Round 2

Reviewer 1 Report

Congratulations to the authors.

I am satisfied with the corrections made to the manuscript.

Author Response

Dear reviewer,

Pleases let me express our appreciations again for your suggestions about this work. We will consider these suggestions in our future research and look forward to receiving your guidance again if possible.

Have a nice day!

Best wishes,

Ms.Ke Shang

Peking University